# Using arterial–venous analysis to characterize cancer metabolic consumption in patients

Nanxiang Xiong [1,9,10✉], Xiaofei Gao[2,9], Hongyang Zhao[1], Feng Cai[2], Fang-cheng Zhang[1], Ye Yuan[1], Weichao Liu[1], Fangping He [3], Lauren G. Zacharias[2], Hong Lin[1], Hieu S. Vu[2], Chao Xing [4], Dong-Xiao Yao[1], Fei Chen[2], Benyan Luo[4], Wenzhi Sun[5,6], Ralph J. DeBerardinis [2,7], Hao Xu[1] & Woo-ping Ge [5,8,10✉]

Understanding tumor metabolism holds the promise of new insights into cancer biology, diagnosis and treatment. To assess human cancer metabolism, here we report a method to collect intra-operative samples of blood from an artery directly upstream and a vein directly downstream of a brain tumor, as well as samples from dorsal pedal veins of the same patients. After performing targeted metabolomic analysis, we characterize the metabolites consumed and produced by gliomas in vivo by comparing the arterial supply and venous drainage. N-acetylornithine, D-glucose, putrescine, and L-acetylcarnitine are consumed in relatively large amounts by gliomas. Conversely, L-glutamine, agmatine, and uridine 5-monophosphate are produced in relatively large amounts by gliomas. Further we verify that D-2-hydroxyglutarate (D-2HG) is high in venous plasma from patients with isocitrate dehydrogenases1 (IDH1) mutations. Through these paired comparisons, we can exclude the interpatient variation that is present in plasma samples usually taken from the cubital vein.

[1] Department of Neurosurgery, Union Hospital, Tongji Medical College, Huazhong University of Science and Technology, 430022 Wuhan, China. [2] Children's Research Institute, University of Texas Southwestern Medical Center, Dallas, TX 75390, USA. [3] Department of Neurology, First Affiliated Hospital, School of Medicine, Zhejiang University, 310003 Hangzhou, China. [4] Eugene McDermott Center for Human Growth and Development, Department of Bioinformatics, Department of Population and Data Sciences, University of Texas Southwestern Medical Center, Dallas, TX 75390, USA. [5] Chinese Institute for Brain Research, Beijing, 102206 Beijing, China. [6] School of Basic Medical Sciences, Capital Medical University, 100069 Beijing, China. [7] Howard Hughes Medical Institute, University of Texas Southwestern Medical Center, Dallas, TX 75390, USA. [8] Children's Research Institute and Department of Neuroscience, University of Texas Southwestern Medical Center, Dallas, TX 75390, USA. [9] These authors contributed equally: Nanxiang Xiong, Xiaofei Gao. [10] These authors jointly supervised this work: Nanxiang Xiong, Woo-ping Ge. ✉email: nanxiangxiong@hust.edu.cn; woopingge@cibr.ac.cn

Gliomas are the most common brain tumors in adults and the most lethal solid cancer in children younger than 12 years old[1,2]. Malignant gliomas remain incurable and present unique challenges for clinicians, radiologists, and translational investigators aiming to improve both diagnosis and prognosis[3]. Targeting tumor metabolism has re-emerged over the last decade as a potential source of new cancer therapies[4]. There are several means by which human gliomas metabolism has been assessed: through the metabolome of plasma collected from the cubital vein, through metabolomics analysis of blood collected from resected cancer tissue or cerebral spinal fluid, through imaging with nuclear magnetic resonance (NMR), and through assessment of isotope enrichment in glioma tissue after intraoperative infusion with $^{13}$C-labeled nutrients[5–10]. To date, however, direct measurement of metabolites consumption and production by gliomas in patients is technically difficult. For example, NMR is limited to a relatively small number of metabolites, e.g., choline, creatine, glutamate, N-acetyl-aspartate (NAA), etc[11].

In this study, we develop a method, named CARVE, paired analysis of Cancer ARterial-VEnous metabolome, that is based on the prediction that gliomas consume metabolites from the arterial blood in appreciable quantities, and that these metabolites are present at significantly lower concentrations in venous blood downstream of the glioma. Conversely, metabolites produced and secreted by gliomas accumulate in venous blood downstream of the glioma relative to the arterial supply. Through the comparison of plasma metabolomes between the arterial supply and venous drainage, we exclude the interpatient variation and characterized multiple metabolites that are consumed and produced by gliomas in vivo from patients.

## Results

**Glioma blood sample collection and targeted metabolomic measurement**. To achieve this goal, we developed a method to collect paired samples of blood upstream and downstream of gliomas from patients (Fig. 1a, b). Shortly before glioma resection, we took a small amount of blood (~1 ml) from an artery and a vein leading directly to and from a glioma, i.e., from arterial and venous locations of glioma vasculature, respectively (Fig. 1c; see details in Methods section). We also collected 1–2 ml blood from the dorsal pedal veins of these patients. We have successfully collected blood samples from these three locations in 13 patients with astrocytoma, oligodendroglioma, glioblastoma (GBM), or gliosarcoma (see Supplementary Table 1). After extracting metabolites from the plasma, we measured 204 metabolites with targeted metabolomic analysis in a liquid chromatograph/triple quadrupole mass spectrometer. We reliably obtained signals from 107 metabolites in each sample.

To identify metabolites consumed and produced by gliomas, we performed arterial–venous comparison from upstream and downstream of the glioma. Glioma arterial and venous metabolite profiles from the same patient tended to cluster together by the unsupervised principal component analysis (PCA) (Fig. 2a, b), indicating that the differences in metabolomes across patients are larger than those between arterial and venous samples from same patient (Fig. 2a, b). Among all of the metabolites that we detected, betaine aldehyde, asymmetric dimethylarginine (ADMA), L-tyrosine, N-acetylornithine, pyruvaldehyde, L-kynurenine, L-phenylalanine, D-glucose, L-methylhistidine, N-alpha-acetyllysine, putrescine, L-acetylcarnitine, L-alanine, and glucosamine were consumed most by gliomas (i.e., Variable Importance in Projection (VIP) score >1 by partial least squares discriminant analysis (PLS-DA) analysis comparing venous and arterial groups). Inosine, hypoxanthine, methionine sulfoxide, succinic acid, adenosine, L-glutamine, choline, myoinositol, L-homoserine, uridine, acetylcholine,

uridine 5-monophosphate, glycerophosphocholine, gamma-aminobutyric acid (GABA), agmatine, lactate, cytidine, taurine, and xanthine were among the metabolites produced by gliomas (Fig. 2c, d). L-Alanine displayed a relative depletion in the venous samples from 13 patients (Fig. 2e) and glucose concentration decreased in most of the samples ($n = 9$ of 13, Fig. 2e). In short, we systemically identified multiple metabolites that are either consumed or produced by gliomas in patients.

**Paired comparison of plasma metabolome**. To determine whether these metabolites are consumed or produced by gliomas, we compared the metabolome of plasma from a glioma artery with that of plasma from the dorsal pedal vein. Because blood in arteries does not pass through the capillary network, metabolite concentrations in arteries (except for the pulmonary arteries) are expected to be similar in different organs. By comparing the metabolomes of dorsal pedal vein and glioma artery plasma, we identified the metabolites consumed or produced by cells in the foot. In this analysis, 20 metabolites including D-glucose and glutamine were reduced in the dorsal pedal vein relative to the artery supplying the glioma. Four of these metabolites, N-acetylornithine, D-glucose, putrescine, and L-acetylcarnitine, were also consumed by gliomas (Fig. 2c, d and Fig. 3a–d). L-glutamine, agmatine, and uridine 5-monophosphate levels were higher in plasma from the glioma vein than in plasma from the dorsal pedal vein, indicating that these metabolites are likely consumed by cells in the foot but produced by gliomas. These results highlight differences in metabolite consumption and secretion among different human organs. Some metabolites produced in one organ (e.g., in the foot: putrescine, agmatine, uridine 5-monophosphate, and xanthine) may feed glioma metabolism in the brain. In nearly all patients, acetylcholine, allantoin, and imidazoleacetic acid were enriched in plasma from the dorsal pedal vein (Fig. 3e). ADP was reduced in plasma from the dorsal pedal vein in all 13 patients, indicating that ADP was consumed by cells in the foot. The metabolomes of the plasma of the dorsal pedal vein and glioma vein could be separated into two groups through PLS-DA analysis, confirming the distinct metabolic profiles of the blood in these two vessels (Fig. 4a). Among these metabolites, L-cystine, L-isoleucine, allantoin, urea, deoxyribose 1-phosphate, imidazoleacetic acid, methionine sulfoxide, adenine, L-methionine, and L-asparagine were found at significantly lower levels in plasma of glioma veins than dorsal pedal veins (Fig. 4b, c). Comparing the metabolomes of plasma from glioma venous samples with those from the dorsal pedal vein, levels of L-glutamine, creatine, 5-aminolevulinic acid, D-glucosamine 6-phosphate, L-3-phenyllactic acid, ADP, riboflavin, agmatine, cis-aconitic acid, adenosine monophosphate, inosinic acid, niacinamide, xanthine, spermidine, cytidine monophosphate, uridine 5-monophosphate, adenosine, S-adenosylmethionine, and hypoxanthine, and inosine were all significantly higher in plasma from glioma veins (Fig. 4b–e).

**2-Hydroxyglutarate (2HG) concentration in glioma arterial and venous plasma**. We found that it was extremely difficult for us to obtain "clean" signals of 2-hydroxyglutarate (2HG) from our measurement with QTRAP because it was always fused with the peak of another unknown metabolite in plasma. They had very close retention times (Supplementary Fig. 1). We observed a tendency that the 2HG concentrations in glioma venous samples from patients with grade II and III gliomas (five of six patients, oligodendroglioma and astrocytoma) were much higher than those in other gliomas (e.g., GBM) after their values were normalized to the samples from a peripheral vein in the same patients (Fig. 5). To further confirm the results, we used 6550

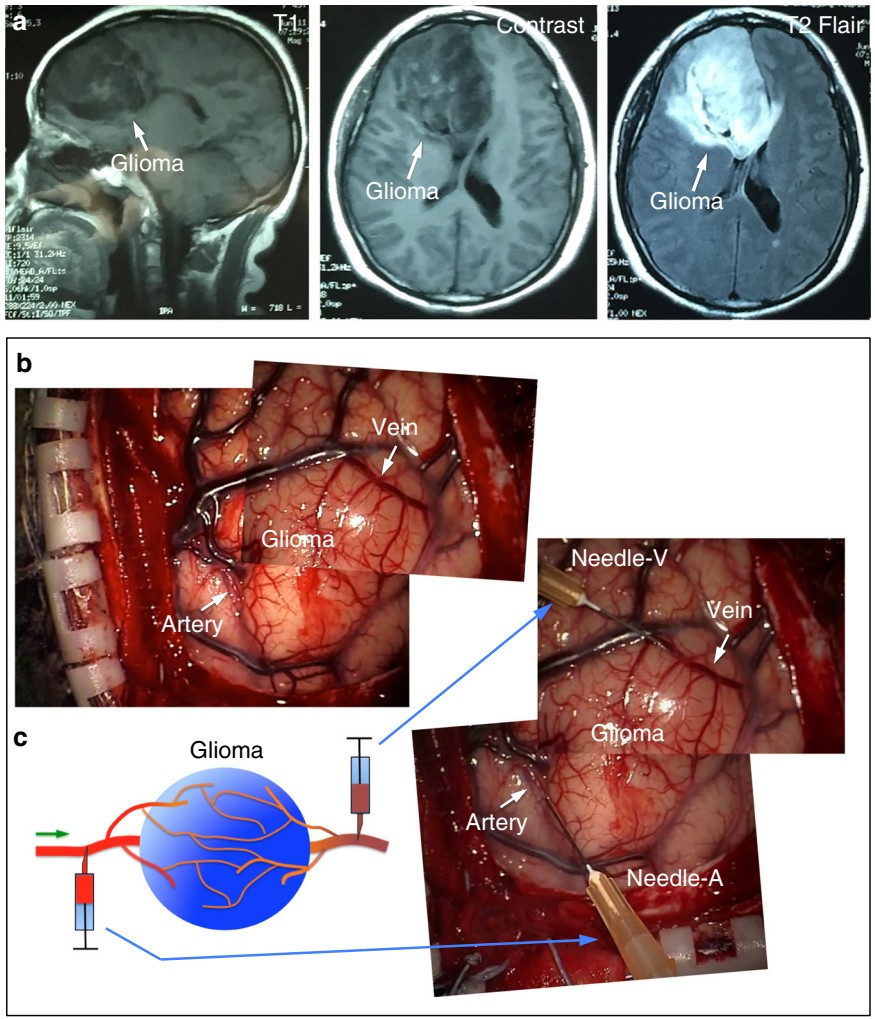

**Fig. 1 The method of blood sample collection from patient glioma arteries and veins. a** MR imaging of brain tumors in patients. The representative images were collected from a patient with a 3.0 T scanner. T2-weighted/fluid attenuated inversion recovery (FLAIR) axial images and postcontrast T1-weighted images were acquired to identify the location of the glioma. **b**, **c** Our strategy for collecting samples of blood from a glioma artery and vein for paired comparison. Left panel, schematic showing the method of glioma blood collection; right panel, combined images showing the locations where blood samples were collected from a patient's glioma. The combined image was generated from two images taken at different time points.

iFunnel Q-TOF LC/MS to measure 2HG from these samples. We observed that 2HG signals in four of six patients with grade II and III gliomas were significantly higher than those in other patients (Supplementary Fig. 2).

Because we did not perform genomic sequencing of gliomas from all of these patients to identify the mutations in these gliomas, we did not know which patient(s) have *IDH1/2* mutations. Somatic mutations in *IDH1* were described in 12% of glioblastomas[12]. *IDH1/2* are commonly mutated genes in grade II and grade III gliomas, with incidences of >75%[13,14]. Fortunately, we had the staining results for some of these patients after surgery (not all glioma samples from the hospital were sent for staining with antibodies against P53, IDH1, and ATRX). Gliomas from four patients had *IDH1* mutations (see Supplementary Table 1). All venous plasma samples from patients with *IDH1* mutations had high 2HG signal (Fig. 5c, Supplementary Fig. 2).

We used a different method[15] to measure D-2HG and L-2HG in samples from these patients with *IDH1* mutations (i.e., patients No. 9–12, Supplementary Table 1). We observed that D-2HG was significantly higher in venous samples compared to arterial samples from the same patients (Fig. 5d). We also noted that the D-2HG concentration in peripheral venous samples was very low

in all peripheral samples (peripheral plasma, $0.67 \pm 0.19$uM; glioma arterial plasma $35.01 \pm 10.31$ uM; glioma venous plasma $48.95 \pm 12.49$ uM, $n = 4$ patients), which is comparable to that of L-2HG concentration ($0.35 \pm 0.04$ μM, $n = 4$ patients). Our results demonstrate that a high amount of D-2HG was released into the blood from gliomas with *IDH1* mutations.

Based on the metabolites enriched in arterial plasma (i.e., consumed by gliomas) and enriched in venous plasma (i.e., they are released from glioma). We did metabolite enrichment analysis. We found that there is largest impact in Phenylalanine, tyrosine and tryptophan metabolism in arterial plasma and purine metabolism pathways in venous plasma (Supplementary Figs. 3 and 4).

## Discussion

The brain consists of multiple cell types that form a complex neuron–glia blood vasculature network. During glioma development, glioma cells infiltrate normal brain tissue and interact with cells in this network[16]. The neighboring non-glioma cells form a unique tumor microenvironment (TME), which is critical for glioma progression[16–18]. It will be of interest to determine whether glioma cells and neighboring non-glioma cells form a metabolic

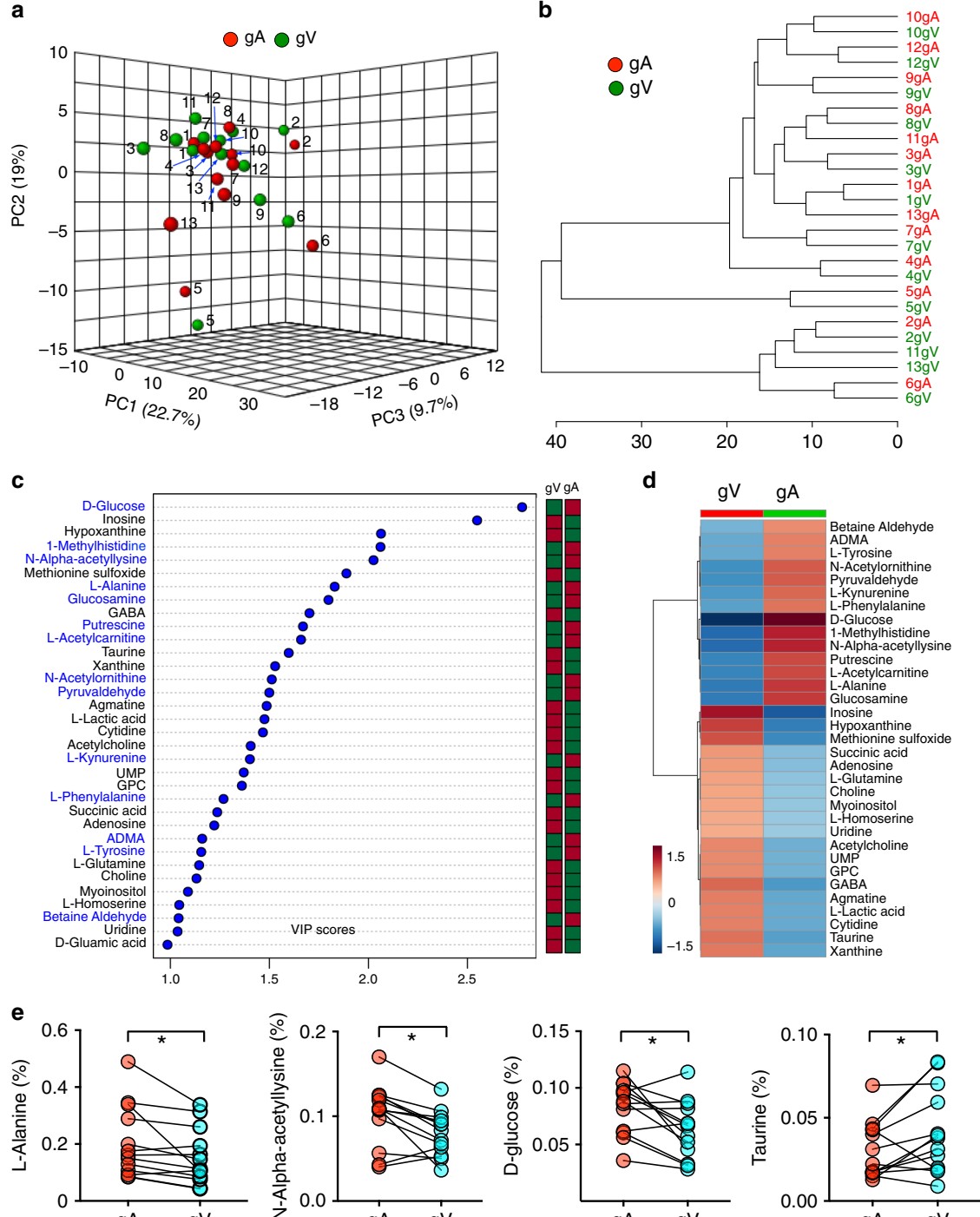

**Fig. 2 Paired comparison of metabolomes from glioma arterial and venous plasma. a** PCA of the 107 metabolomes of glioma arterial and glioma venous samples from 13 patients. The top three PCs explain 51.4% (19%, 22.7%, and 9.7%) of the total variance. Each circles indicate an individual sample of patient arterial or venous blood. gA (samples from glioma arteries, red) and gV (samples from glioma veins, green) cannot be separately by the unsupervised PCA. **b** Clustering analysis of the metabolomic data from 13 patients. **c** VIP scores of metabolites by PLS-DA. The names of metabolites enriched in glioma arterial plasma (gA) are labeled blue. The columns to the right indicate whether the abundance of each metabolite is enhanced (red) or reduced (green) in the arterial or venous plasma group. **d** Heatmap of metabolites levels with VIP score > 1. Color bar (bottom left) indicates the scale of standardized metabolite levels. Warm color indicates higher concentration. **e** Relative abundance (normalized by TIC, Total Ion Chromatogram, y-axis) of four representative metabolites (L-alanine, N-alpha-acetyllysine, D-glucose, and Taurine) from arterial (gA) and venous (gV) samples. *p < 0.05; two-tailed paired t-test. GPC, Glycerophosphocholine. GABA, Gamma-Aminobutyric acid. UMP, Uridine 5-monophosphate.

ecosystem to support each other. In our current study, we cannot exclude the contribution of metabolites produced by non-glioma cells. The extent of the contribution of these non-glioma cells to the glioma metabolome that we measured from glioma plasma is

unknown and difficult to answer. Comparing the metabolomes of arterial and venous plasma from the same patient is an efficient method to exclude the large variations observed across patients (Figs. 2a, e, 3a, e, 4a, e). Our strategy greatly increases the chance of

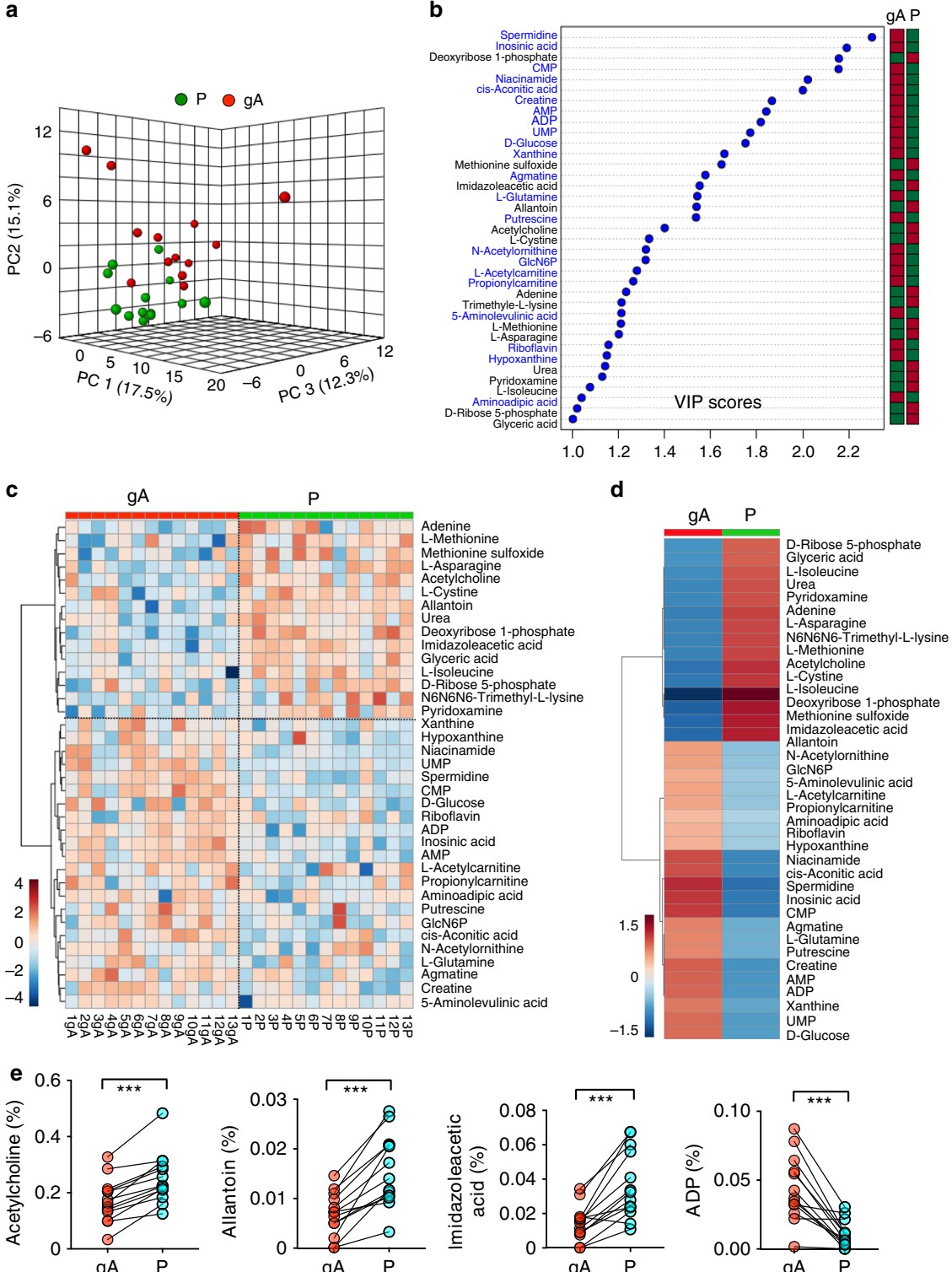

**Fig. 3 Paired comparison of metabolomes from glioma artery and dorsal pedal vein plasma. a** PCA of the metabolomes of glioma artery (gA) and dorsal pedal vein (P) samples from 13 patients. Three PCs explain 44.9% (15.1%, 17.5%, and 12.3%) of variance between gA and P. PC scores are indicated as %; circles indicate individual samples from the glioma artery and dorsal pedal vein. **b** VIP scores of metabolites between glioma artery(gA) and dorsal pedal vein (P) plasma samples. The names of the metabolites enriched in glioma arterial blood (gA) are labeled with blue text. The columns to the right indicate whether the abundance of each metabolite is enhanced (red box) or reduced (green box) in each plasma group. **c** Heatmap representation of 37 of 107 metabolites of blood plasma from glioma artery (gA) and dorsal pedal vein (P) samples. Warm color indicates higher concentration. **d** A heatmap representation of 37 metabolites (VIP score >1) in blood samples from glioma artery (gA) and dorsal pedal vein (P). All metabolites are listed on the right side of the map. Color bar (bottom left), warm color indicates higher concenration. **e** Relative abundance (normalized by TIC, *y*-axis) of four representative metabolites (Acetylcholine, Allantoin, Imidazoleacetic acid, ADP) from glioma artery (gA) and dorsal pedal vein (P) samples. ***$p < 0.001$; two-tailed paired *t*-test. GlcN6P, Glucosamine 6-phosphate. CMP, Cytidine monophosphate. AMP, Adenosine monophosphate. ADP, Adenosine diphosphate.

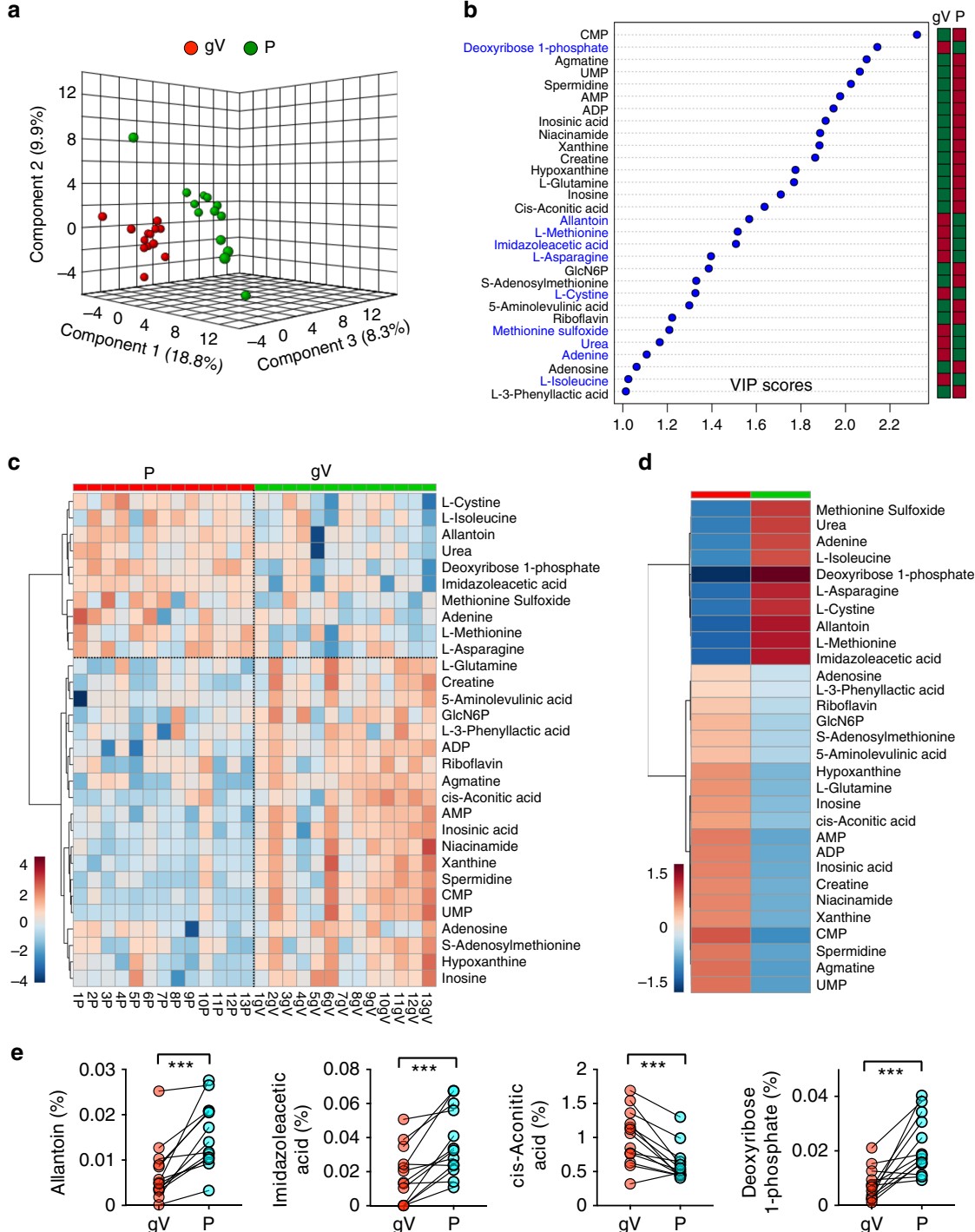

**Fig. 4 Paired comparison of metabolomes from glioma vein and dorsal pedal vein plasma. a** PCA of the metabolomes of glioma vein (gV) and dorsal pedal vein (P) samples from 13 patients. Three PCs explain 36.9% (9.8%, 18.3%, and 8.8%) of variance and could not separate gV from P. PC scores are indicated as %; circles indicate individual samples from glioma venous blood and dorsal pedal vein samples, respectively. **b** VIP scores of metabolites between glioma vein and dorsal pedal vein plasma samples. The metabolites enriched in glioma venous blood (gV) are labeled in blue. The columns to the right indicate whether the abundance of each metabolite is enhanced (red box) or reduced (green box) in each plasma group. **c** A heatmap representation of 30 of 107 metabolites of blood plasma from gV and P samples of 13 patients. Warm color indicates higher concentration. **d** A heatmap representation of 30 metabolites (VIP score > 1) in blood samples from the glioma vein (gV) and dorsal pedal vein (P). Names of these metabolites are listed on the right side of the map. Color bar (bottom left), warm color indicates higher concentration. **e** Relative abundance (normalized by TIC, y-axis) of four representative metabolites (Allantoin, Imidazoleacetic acid, cis-Acontic acid, Deoxyribosoe 1-phosphate) from glioma vein (gV) and dorsal pedal vein (P) samples. ***p < 0.001; two-tailed paired t-test.

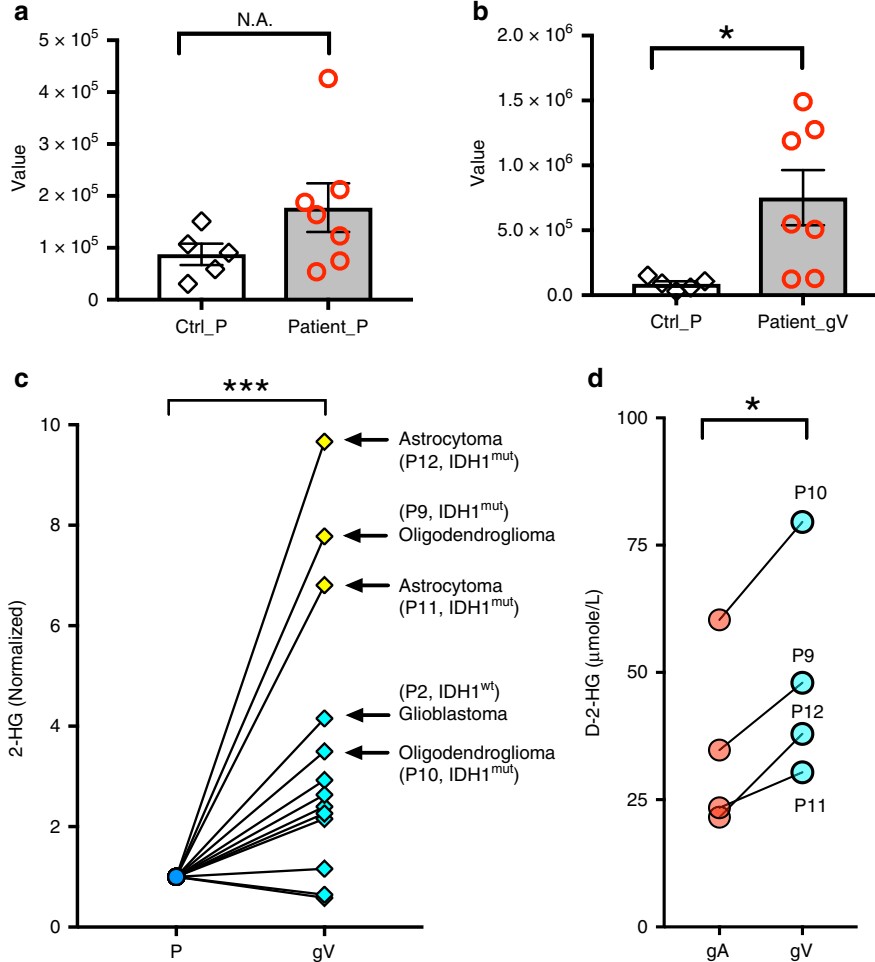

**Fig. 5 Measurement of 2HG from human blood plasma. a–c** The 2HG concentrations in glioma venous samples from patients with grade II and III gliomas were much higher than those with other gliomas (e.g., GBM). In **a** and **b**, Ctrl, $n = 5$; patients with oligodendroglioma and astrocytoma, $n = 7$, in **c**, all patients in Supplementary Table 1, $n = 13$. **d** Measurement of D-2HG in arterial, venous, and peripheral samples from these patients with *IDH1* mutations (based on staining results, patients No. 9–12, also see Supplementary Table 1). D-2HG was significantly higher in venous samples than in arterial samples from the same patients. Ctrl_P, plasma from dorsal pedal vein of control subjects. Patient_gV, plasma from glioma veins of patients. gA samples from glioma arteries, gV samples from glioma veins, P samples from dorsal pedal vein. ***$p < 0.001$, *$p < 0.05$. Two-tailed non-paired *t*-test for **a**, **b**. Two-tailed paired *t*-test for **c** and **d**. All data were presented as mean ± sem.

identifying metabolites consumed or produced by gliomas, which are impossible to detect in blood samples from the dorsal pedal vein or cubital vein, where blood samples have traditionally been collected for metabolomic analysis.

It has been reported that some metabolites are higher in the cerebrospinal fluid of glioma patients than in control subjects, including taurine, hypothanine, and L-glutamine[5]. Consistent with these observations, we also detected that these metabolites, relative to glioma arteries, are present at higher levels in plasma collected from glioma veins. It is therefore likely that gliomas produce these metabolites. Currently, increasing numbers of metabolites have been identified using NMR for brain tumor diagnosis, as these tests are inexpensive and can be done within a short time[19]. Gliomas exhibit markedly different spectra from those of neighboring normal brain tissue[20,21]. When the metabolic ratios of choline (Cho), N-acetyl-aspartate (NAA) and creatine are assessed in the spectra via chemical shift imaging[22,23], nearly all gliomas are found to have decreased NAA and increased choline, thus producing an abnormally high Cho/NAA ratio in glioma tissue. The decrease in NAA is widely interpreted as a sign of neuronal loss or damage[24,25], and increased choline is often thought to represent the dramatic increase of membrane

synthesis in proliferating glioma cells[26]. Interestingly, we also found that choline is produced by gliomas (low in glioma arterial plasma but higher in glioma venous plasma).

We did not detect high D-2HG in peripheral venous plasma, which is consistent with the results from a previous study of D-2HG in peripheral venous samples[27]. However, although significantly lower than those in venous samples, we surprisingly detected that D-2HG levels were also high in glioma arterial plasma compared to peripheral plasma. This is likely because the glioma arterial vessel from which we collected blood in the cerebral cortex was located right above the glioma (due to ethical issues, this is a safe location that we were allowed for blood collection). Some amount of D-2HG might pass through the endothelial cells and enter the glioma arterial vessel. The method that we developed here allows us to successfully collect from an artery and vein specifically upstream and downstream of a brain tumor in patients for the purpose of performing metabolomic analysis to characterize the uptake and consumption of metabolites from the tumor. Identification of the metabolites consumed by gliomas in vivo is beneficial for the understanding of glioma metabolism. Our results may also provide clues for researchers to develop imaging biomarkers for

distinguishing various glioma subtypes or evaluating the progression of gliomas.

We demonstrated the feasibility of metabolomic comparison of arterial–venous samples from patients with brain tumors. It will encourage the scientific community to use a similar strategy to perform metabolomic analysis of other cancers in patients or animal models. Identification of the metabolites or other molecules consumed by gliomas in vivo is beneficial for the understanding of glioma metabolism. However, more follow-up experiments need to be performed in the future. In addition, the metabolites from arterial and venous samples might not be directly derived from cancer cells. They might be intermediate metabolites from the tumor microenvironment. The metabolites may be released from both cancer and noncancer cells. Also, we cannot exclude the possibility that brain vascular cells located in the arterial and venous segments contribute to the difference in metabolite concentrations.

## Methods

**Patient selection**. All patients were enrolled in the study approved by the Institutional Review Board at Tongji Medical College, Huazhong University of Science and Technology (IRB, 2017-S229). Informed consent was obtained from all patients. The gender and age information of human subjects were included in the data tables for plasma metabolomics. We ensured that data users or all people involved in this study agree to protect participant confidentiality when handling data that contains potentially identifying information from all patients. The patients ($n = 27$) included both males and females with an age range of 21–61 years old. The average age was $46.8 \pm 2.2$ years old. Samples from 13 patients were recovered without hemolysis. All of these 13 patients were diagnosed with supratentorial glioma including five cases of astrocytoma, two cases of oligodendroglioma, five cases of glioblastoma, and one case of gliosarcoma (see the detailed information on these 13 patients in Supplementary Table 1). The procedure was tolerated in an additional 14 patients in whom excessive hemolysis precluded metabolomics.

**Immunohistochemistry analysis**. We used tissue sections for routine histological examination and for immunohistochemistry. These sections contained regions for glioma diagnosis. Tumor tissues were fixed in formalin and then embedded in paraffin. Sections were cut at $6 \mu m$. Slides were then incubated with antibodies against IDH1 R132H (1:300, clone H09, ZM0447, ZSGB-Bio), ATRX (1:200, ZA-0016, polyclonal, ZSGB-Bio), and P53 (1:300, DO-7, mouse monoclonal, Roche) for staining after surgery.

**Blood collection from glioma blood vessels and peripheral vein**. Patients were typically in the supine position. After tracheal intubation, the patient's head was fixed to a May-field frame and tilted to the left or right, dependent on the location of the tumor. A solution of 1% iodophor was applied to disinfect the incision. After cutting the epidermis and drilling the skull, the skull was milled to create a bone flap. The dura matter was cut with horseshoe scissors to reveal the glioma. The texture and color of glioma tissue is obviously different from that of normal brain tissue. Under the operating microscope, the arteries and veins of glioma tissue could be clearly identified. During resection, both the arteries and veins of gliomas need to be coagulated and cut. We collected 1 ml blood with a syringe (1 ml LS 25GA, 5/8 inch, BD) from an upstream glioma artery and downstream glioma vein of each patient. Blood can be smoothly withdrawn from the arterial and venous vessels of gliomas. We also collected 1–2 ml of peripheral limb venous blood from the dorsal pedal vein. Thus, we obtained three blood samples from each patient.

**Blood sample preparation after collection**. Fasting blood samples (1 ml) were collected as described above from patients before glioma resection surgery. These samples were collected in tubes with anticoagulant heperin and stored on ice. They were then shipped to a laboratory for centrifuging. Samples were immediately placed on ice for 15 min. Then, the samples were centrifuged in the laboratory for 5 min ($1000 \times g$, 4 °C). Samples were evaluated for hemolysis, and only samples without hemolysis were analyzed. Each sample of nonhemolytic plasma was divided into three aliquots of $100 \mu l$ each for metabolomic experiments. All aliquots were stored at $-80$ °C before metabolite extraction.

**Purification of metabolites from blood**. Plasma samples were thawed at 4 °C, and $100 \mu l$ of plasma was collected into an Eppendorf tube containing $900 \mu l$ of ice-cold methanol/80% water (vol/vol) (pre-cooled at $-80$ °C) (V plasma: V methanol = 1:9). After being concussed rigorously for 1 min, the mixture was prepared by centrifugation ($17,000g$ 15 min) in a refrigerated centrifuge. Then, $800 \mu L$ of the metabolite-containing supernatant was transferred to a new Eppendorf tube, and the protein pellet was collected for protein quantitation. $100 \mu l$ of supernatant (i.e.,

metabolites from $\sim 12.5 \mu l$ blood) was dried in a SpeedVac at room temperature to obtain a pellet, which was stored at $-80$ °C before performing metabolomic profiling analysis.

**Analysis of metabolites from blood plasma**. For targeted metabolomic analysis, metabolites in blood plasma were reconstituted in $50 \mu l$ of 0.03% formic acid in water and then analyzed with a SCIEX QTRAP 5500 liquid chromatograph/triple quadrupole mass spectrometer. Using a Nexera Ultra-High-Performance Liquid Chromatograph system (Shimadzu Corporation), we achieved the separation on a Phenomenex Synergi Polar-RP HPLC column ($150 \times 2$ mm, $4 \mu m$, 80 Å). The mass spectrometer was used with an electrospray ionization (ESI) source in multiple reaction monitoring (MRM) mode[28]. We set the flow rate with 0.5 ml/min, and the injection volume with $20 \mu l$. We acquired MRM data with Analyst 1.6.3 software (SCIEX).

In Supplementary Fig. 2, we performed non-targeted metabolomic analysis of plasma samples (including 2HG measurement) in a 1290 UHPLC liquid chromatography (LC) system interfaced to a high-resolution mass spectrometry (HRMS) 6550 iFunnel Q-TOF mass spectrometer (MS) (Agilent). Both positive and negative (ESI+ and ESI-) modes were used. Analytes were separated on an Acquity UPLC® HSS T3 column ($1.8 \mu m$, $2.1 \times 150$ mm, Waters). Mobile phase A composition was 0.1% formic acid in water and mobile phase B composition was 0.1% formic acid in 100% ACN. ESI source conditions were set as follows: dry gas temperature 225 °C and flow 18l/min, fragmentor voltage 175 V, sheath gas temperature 350 °C and flow 12l/ min, nozzle voltage 500 V, and capillary voltage $+3500$ V in positive mode and $-3500$ V in negative. Raw data files were processed using Profinder B.08.00 SP3 software (Agilent).

**Measurement of D-2HG from plasma**. In the measurement of Fig. 5d, metabolites were extracted with 80% methanol–water solution from $25 \mu l$ plasma from patients in a tube. A SpeedVac was used to dry the extract into a pellet. To the pellet was added U13C-D/L-2HG (internal standard, Cambridge isotope laboratories, 10 nG in $10 \mu l$ acetonitrile), then the mixture was then dissolved in $90 \mu l$ freshly mixed 80% acetonitrile/20% acetic acid plus 50 mG/ml diacetyl-L-tartaric anhydride (DATAN, Acros Organics). The solution thus obtained was sonicated and warmed up to 75 °C for 30 min. Samples were cooled to room temperature and centrifuged. The supernatant was dried with a SpeedVac, and the pellet was reconstituted into 1.5 mM ammonium formate aqueous solution with 10% acetonitrile ($100 \mu l$). LC/MS analysis was performed on an AB Sciex 5500 QTRAP liquid chromatography/mass spectrometer (Applied Biosystems SCIEX) equipped with a vacuum degasser, a quaternary pump, an autosampler, a thermostatted column compartment, and a triple quadrupole/iontrap mass spectrometer with electrospray ionization interface, and controlled by AB Sciex Analyst 1.6.1 Software. Waters Acquity UPLC HSS T3 column ($150 \times 2.1$ mm, $1.8 \mu M$) column was used for separation. Solvents for the mobile phase were 1.5 mM ammonium formate aqueous (pH 3.6 adjusted with formic acid (A), and pure acetonitrile (B). The gradient elution was: 0–12 min, linear gradient 1–8% B and 12–15 min, 99% B, then the column was washed with 99% B for 5 min before reconditioning it for 3 min using 1% B. The flow rate was 0.25 ml/min and the column was operated at 35 °C. Multiple reaction monitoring (MRM) was used to check 2-hydroxyglutarate-diacetyl tartrate derivatives: 363/147 (CE: $-14$V); 368/152 (internal standard, CE: $-14$V).

**Data analysis**. Integrated chromatogram peaks of each metabolite were analyzed with MultiQuant software (AB Sciex). The ion intensity was calculated by normalizing single ion values against the total ion value of the entire chromatogram (i.e., TIC or Total Ion Chromatogram). The data matrix was input into SIMCA-P software (Umetrics) by mean-centering and Pareto scaling for subsequent analysis so that the model fitting would not be biased by concentrations and variations of different metabolites. Both unsupervised and supervised multivariate data analysis approaches including PCA, hierarchical clustering, and PLS-DA were performed using Metaboanalyst 4.0[29] and then plotted with Prism 7.0. We performed feature selection in PLS-DA to identify metabolites that maximize separation between the venous and arterial groups by rotating the PCA components. The importance of a metabolite in the model is measured by the VIP score. The VIP score of a metabolite is calculated as a weighted sum of the squared correlations between this metabolite and the derived PLS-DA components. Each weight corresponds to the percentage variation of the response variable, i.e., gA and gV, explained by a PLS-DA component. Intuitively, the VIP score of a metabolite indicates its intensity of association with the PLS-DA components that best distinguish the gA and gV groups. By definition, the average of squared VIP scores equals 1, and by convention a VIP score of greater than 1 is used as a criterion for variable selection[30]. Thus, metabolites with a VIP score > 1 were reported. All data were presented as mean ± sem.

**Magnetic resonance imaging of brain tumors in patients**. All patients were imaged in a clinical 3.0 T scanner (Magnetom Verio, Siemens Healthcare, Erlangen, Germany) equipped with a 12-channel head coil using T1-weighted coronal and axial imaging and T2-weighted/FLAIR axial imaging. Postcontrast T1-weighted images were acquired after injection of either gadopentetate dimeglumine (Magnevist, Bayer Schering Pharma AG) or gadobenate dimeglumine (Multihance, BD),

administered at a dose of 0.1 mmol/kg. The MR imaging protocol was as follows: T1-weighted images were acquired using an echo time (TE) 2.48 ms, a repetition time (TR) 300 ms, and voxel size $0.9 \times 0.7 \times 6$ mm. T2-weighted FLAIR images were acquired using an inversion time of 2500 ms, TR 9000 ms, TE 90 ms, and voxel size of $0.9 \times 0.9 \times 6$ mm.

**Reporting summary**. Further information on research design is available in the Nature Research Reporting Summary linked to this article.

## Data availability

The data that support the findings of this study are available in the Article file, Supplementary Information or available from the corresponding author upon reasonable request. Source data underlying the Figs. 2–5 and Supplementary Fig. 2 are available as a Source Data file. Source data are provided with this paper.

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

## Acknowledgements

We sincerely appreciate the support from patients. We thank Dr. F. Huang for advice in data analysis, B. Samuels for critical reading of the manuscript, Dr. B. Mickey from Dept. of Neurological Surgery at UTSW and Dr. X.M. Liu from HUST for critical discussion. We thank J. L. Li, Z.P. Hu, and other colleagues from CIBR or CRI for discussion on the work. This work was supported by CIBR startup funds, and Jonesville foundation (for Brain Tumor Research) to W.G.; National Natural Science Foundation of China (No.81671210, 81371380) and National Key Technology Research and Development Program of the Ministry of Science and Technology of China (2014BAI04B01, 2013BAI09B03) to N.X., and National Cancer Institute R35CA22044901 to R.J.D.

## Author contributions

W.G. and N.X. conceived the project. W.G., N.X., X.G., F. Cai, and H.S.V. designed the experiments. W.G., F.H., and X.F. extracted metabolites. X.G and W.G. completed pilot experiments. F. Cai performed D-2HG analysis. W.G., F. Chen, L.Z., H.S.V., F. Cai, and C.X. completed metabolomics analysis. H.Z., F.Z., H.L., D.Y., H.X., and N.X. performed sample collection from patients, Y.Y., W.L., and H.X. prepared samples and sample storage, W.G., X.N., W.S., B.L., and R.J.D. provide reagents. W.G. and X.N. wrote the manuscript. All authors discussed, reviewed, and edited the manuscript.

## Competing interests

The authors declare no competing interests.
