## [Peer Review File · Nature Communications]

Reviewers' comments:

Reviewer #1 (Remarks to the Author): Expert in metabolomics

In this manuscript entitled "Using arterial-venous analysis to characterize cancer metabolic consumption in patients", Xiong and colleagues performed metabolomics analyses of the blood from arteries and veins surrounding brain tumours in humans. They compare these analyses to peripheral blood in the foot.

This report is to my knowledge the first to present arterial-venous metabolomics analyses in human patients. However, apart from this aspect, the paper falls short in providing any biological or clinical relevance, critically reducing the impact of the work. Also, several of the metabolites presented in the heatmaps and VIP score graph do not have established biological functions, further limiting the relevance of the findings. In its current form, this work appears more appropriate for a more specialised journal.

Reviewer #2 (Remarks to the Author): Expert in metabolomics

The manuscript "Using arterial-venous analysis to characterize cancer metabolic consumption in patients" provides blood metabolome data of artery and vein around human glioma, which is rare and highly valuable for cancer metabolism. Some quantitative methods like principle components analysis (PCA) are utilized to cluster the data. This study shows those metabolites uniquely produced and consumed by gliomas. Generally this manuscript provides some informative results and will be valuable to the cancer metabolism community. Nevertheless, some questions in data processing and interpretation need to be addressed. Most importantly more mathematical rigor is needed especially regarding the definition of the VIP scores.

(0) The abstract needs much more detail.

(1) The variation of metabolites are reflected by VIP score. However, there is no rigorous definition nor clear explanation of VIP score. In Methods part, only "VIP scores > 1 between groups were considered as significantly discriminating" is mentioned. However, what does VIP scores reflect? Is it non-negative? Is it dimensionless? Will it bias some high-concentration metabolite (such as glucose)? As a metric of variation, those properties may largely affect its reliability and applicability, and therefore should be explained in detail to readers.

(2) One of the main conclusions is unique metabolites produced or consumed by glioma compared to pedal tissue. Therefore, a rigorous definition of "unique" should be introduced. For instance, supplement figure 2 displays VIP scores between glioma vein and pedal vein, but requires a intuitive explanation for those statistics. Readers need to know what's the clear criteria of "unique metabolites".

(3) Some confusing points are in figure 2: figure 2C shows each metabolite has one VIP score, and they are all positive, but figure 2D shows each metabolite has two VIP scores for artery and vein, and some of them are negative. Their scales are also different. I guess it is because PCA is executed on different data set. In 2C PCA is done in total data set and in 2D PCA is done separately in artery and vein for each metabolite. These different protocols on data processing should be clearly explained to readers. Furthermore, two columns in heatmap in figure 2D seems to be exactly opposite. The meaning of this subfigure requires more explanation.

(4) Some confusing points are in supplement figure 1 and 2: similar with comment #(3), they have different VIP scores in different subplots. Furthermore, heatmaps in supplement figure 1D and 2D look very strange. Those heatmaps seem to be binary. What information do they express?

Reviewer #3 (Remarks to the Author): Expert in glioma surgery

In their manuscript entitled "Using arterial-venous analysis to characterize cancer metabolic consumption in patients" Xiong and colleagues provide insight into the metabolic properties of gliomas through a comparison of metabolites within arterial and venous blood associated with human gliomas. I have been a reviewer for over one hundred manuscripts in my career. This is easily one of the most interesting and bold experimental designs I have encountered. I have a strong bias towards the publication of this paper for this reason alone. I think the data are extraordinarily valuable because of the rigor of the experimental design and they are hard to argue with. I can think of no better way to study the key elements of human glioma metabolism. Therefore, I think the results will be of high value to the field.

In glioma biology and in the care of glioma patients the IDH mutation is central. I was surprised in reading this manuscript- which certainly contains IDH mutant and wild-type tumors that the authors did not evaluate the upstream and downstream concentrations of 2HG. I view this as an essential internal control for their methodology. As there is no physiologic source of 2HG in the body, 2HG should be low in arterial blood and high in venous blood in an IDH mutant tumor. In IDH wild-type tumors the concentration of 2HG should not vary between arterial and venous blood- and may not be detectable. To me, including a detailed analysis of 2HG in the collected samples is essential. Without this, I think the authors are missing a key control. With 2HG measurements that make sense, I think the authors will greatly improve the acceptance of the other metabolic results they provide.

Manuscript Number: NCOMMS-19-26465-T

Using arterial-venous analysis to characterize cancer metabolic consumption in patients

A summary of our revision:

We have rewritten the abstract, added more details in the Discussion and Methods section, and provided a more detailed description and presentation of the VIP results. In addition, to address the concerns of the reviewers, we used QTOF to measure 2HG from these samples. We have also added several new figures including **Supplementary Figures S3, S4, S5, S6 and S7** in the revised manuscript. Through addressing the reviewers' concerns, we believe the quality of the revised manuscript has been significantly improved.

Reviewers' comments:

Reviewer #1 (Remarks to the Author): Expert in metabolomics

In this manuscript entitled "Using arterial-venous analysis to characterize cancer metabolic consumption in patients", Xiong and colleagues performed metabolomics analyses of the blood from arteries and veins surrounding brain tumours in humans. They compare these analyses to peripheral blood in the foot.

This report is to my knowledge the first to present arterial-venous metabolomics analyses in human patients. However, apart from this aspect, the paper falls short in providing any biological or clinical relevance, critically reducing the impact of the work.

Response

We truly appreciate the reviewer's comments on our work. In our revised manuscript, we performed more experiments to assess 2HG in these samples with QTOF. In addition, we have added more discussion in the revised version regarding the biological significance of our results. We did not perform further mechanistic studies of glioma metabolism in patients because it is extremely challenging for us to go further in patients at this stage. However, our results may encourage other researchers with different expertise from the cancer research community to continue further mechanistic investigation.

(1) We have added more discussion in our revised version to address the reviewer's concern. We have discussed the weakness of our study (highlighted in the main text). The significance of this study lies in demonstrating the feasibility of metabolomic comparison of arterial-venous samples from patients with brain tumors. It will encourage the scientific community to use a similar strategy to perform metabolomic analysis of other cancers in patients or animal models. Identification of the metabolites or other molecules consumed by gliomas *in vivo* is beneficial for the understanding of glioma metabolism. However, more follow-up experiments need to be performed in the future. In addition, the metabolites from arterial and venous samples might not be directly derived from cancer cells.

They might be intermediate metabolites from the tumor microenvironment. The metabolites may be released from both cancer and non-cancer cells. Also, we cannot exclude the possibility that brain vascular cells located in the arterial and venous segments contribute to the difference in metabolite concentrations.

(2) To further confirm the results, we used QTOF (Agilent 6550 iFunnel LC/quadrupole-time of flight mass spectrometer) to measure 2HG from these samples. We observed that 2HG signals in 4 of 6 patients with grade II and III gliomas were significantly higher than in other patients (**Supplementary Figure 5**). It is a pity that we did not perform genomic sequencing of the gliomas from all of these patients and identify the mutations in these gliomas, so we don't know which patient(s) may have an *IDH1/2* mutation. Somatic mutations in isocitrate dehydrogenases1 (*IDH1*) were first described in 2008 as occurring in 12% of glioblastomas (Parsons et al, 2008). *IDH1/2* are the commonly mutated genes in grade II and grade III gliomas, with incidences of >75% (Kim et al., 2010, Hartmann et al., 2009). Our analysis of 2HG, although not ideal, reasonably reflects the metabolism of grade II and III gliomas in patients. We have included all of these results in **Supplementary Figure 3–5** in our revised manuscript.

Also, several of the metabolites presented in the heatmaps and VIP score graph do not have established biological functions, further limiting the relevance of the findings. In its current form, this work appears more appropriate for a more specialised journal.

Response

Based on the metabolites enriched in arterial plasma (i.e., consumed by gliomas) and enriched in venous plasma (i.e., they are released from glioma). We did metabolite enrichment analysis. We found that there is largest impact in Phenylalanine, tyrosine and tryptophan metabolism and purine metabolism pathways in arterial plasma and venous plasma respectively (see the results below. These results are now included in the revised manuscript as Supplementary figure s6 and s7, also see them below).

Supplementary figure S6 (consumed by glioma)

Supplementary figure S7 (released from glioma)

Reviewer #2 (Remarks to the Author): Expert in metabolomics

The manuscript “Using arterial-venous analysis to characterize cancer metabolic consumption in patients” provides blood metabolome data of artery and vein around human glioma, which is rare and highly valuable for cancer metabolism. Some quantitative methods like principle components analysis (PCA) are utilized to cluster the data. This study shows those metabolites uniquely produced and consumed by gliomas. Generally this manuscript provides some informative results and will be valuable to the cancer metabolism community. Nevertheless, some questions in data processing and interpretation need to be addressed. Most importantly more mathematical rigor is needed especially regarding the definition of the VIP scores.

Response:

We thank the reviewer for this encouraging feedback about our study.

(0) The abstract needs much more detail.

Response:

As the reviewer suggested, we have added more details in the abstract (see below).

Abstract (revised)

Understanding tumor metabolism holds the promise of new insights into cancer biology, diagnosis and treatment. To assess human cancer metabolism, we developed a novel method to collect intra-operative samples of blood from an artery directly upstream and a vein directly downstream of a brain tumor, as well as samples from dorsal pedal veins of the same patients. After performing targeted metabolomic analysis in a liquid chromatograph/triple quadrupole mass spectrometer, we reliably obtained signals from over 100 metabolites in each sample. We have characterized the metabolites that are consumed and produced by gliomas *in vivo* by comparing the arterial supply and venous drainage. Our studies demonstrate that four of these metabolites, N-acetylmethionine, D-glucose, putrescine, and L-acetylcarnitine, were consumed in relatively large amounts by gliomas. Conversely, L-glutamine, agmatine, and uridine 5-monophosphate were produced in relatively large amounts by the glioma mass. Through paired comparisons of upstream and downstream samples from the same patient, we are able to exclude the variation across patients that is present in plasma samples usually taken from the cubital vein.

(1) The variation of metabolites are reflected by VIP score. However, there is no rigorous definition nor clear explanation of VIP score. In Methods part, only “VIP scores > 1 between groups were considered as significantly discriminating” is mentioned. However, what does VIP scores reflect? Is it non-negative?

Is it dimensionless? Will it bias some high-concentration metabolite (such as glucose)? As a metric of variation, those properties may largely affect its reliability and applicability, and therefore should be explained in detail to readers.

Response:

We apologize for not including much detail about how we obtained the VIP score in the original manuscript. In our revision, we have added the following information with considerably more detail in the Method section.

- 1) We have clarified in the “Metabolites data analysis section” that the data matrix was first input into SIMCA-P software (Umetrics) by mean-centering and Pareto scaling so that the subsequent analysis and model fitting would not be biased by concentrations and variations of different metabolites.
- 2) We now added more details on the partial least squares discriminant analysis (PLS-DA) method for better and easier interpretation by the audience. Specifically, we clearly state the definition of VIP as “the weighted sum of squares of the PLS loadings for that variable”—because it is a sum of squares, it is a non-negative scalar. As a convention, we report metabolites with VIP score > 1 .
- 3) We have also added the following information to the Method section regarding data analysis:
We used supervised multidimensional statistical methods, specifically partial least square discriminant analysis (PLS-DA), to analyze results from the arterial and venous groups and obtain information about metabolites that differed significantly between the gA and gV groups. ‘Variable importance in the projection’ (VIP) scores were computed and Welch’s *t*-test was applied to determine discriminatory variables in the dataset. The VIP is an estimation of the importance of each variable in the projection used in the PLS-DA model as a quantitative estimation of the discriminatory power of each individual feature. Variables with a VIP score of ≥ 1 were considered important in the PLS-DA model.

(2) One of the main conclusions is unique metabolites produced or consumed by glioma compared to pedal tissue. Therefore, a rigorous definition of “unique” should be introduced. For instance, supplement figure 2 displays VIP scores between glioma vein and pedal vein, but requires a intuitive explanation for those statistics. Readers need to know what’s the clear criteria of “unique metabolites”.

Response:

We thank the reviewer for pointing out that *unique* is not an accurate word here. We have rewritten this part and replaced “unique metabolite consumed by glioma” with “metabolites consumed by glioma”

which we think is a more appropriate description of our results. We have also rewritten other sentences with the word *unique* in the manuscript.

(3) Some confusing points are in figure 2: figure 2C shows each metabolite has one VIP score, and they are all positive, but figure 2D shows each metabolite has two VIP scores for artery and vein, and some of them are negative. Their scales are also different. I guess it is because PCA is executed on different data set. In 2C PCA is done in total data set and in 2D PCA is done separately in artery and vein for each metabolite. These different protocols on data processing should be clearly explained to readers. Furthermore, two columns in heatmap in figure 2D seems to be exactly opposite. The meaning of this subfigure requires more explanation.

Response:

We apologize for the confusing information. We have added more details in Figure 2C.

The heatmap in Figure 2C provides intuitive visualization of a data table for gA and gV samples. Each colored cell on the map corresponds to a concentration value in our data table, with samples in rows and metabolites in columns. We used this heatmap to identify samples/features that are unusually high/low. We only plot the data from the metabolites which show $VIP > 1$ (33 metabolites) from Figure 2B. Data from each sample has been normalized. The values were subjected to auto scaling (mean-centered and divided by the standard deviation of each variable) and presented after Log transformation (generalized logarithm transformation). The distance measurement was Euclidean. The clustering algorithm used was Ward. Each colored box stands for the average value of one metabolite from these normalized data from all gA samples (A) or all gV samples (V). Because the value from each sample has undergone log transformation, the colors vary from deep blue to dark brown to indicate data values (note that this is the TIC, not the VIP score) ranging from very low (cold) to extremely high (hot). We have added all of this information in our Data Analysis (see the highlighted paragraph)

(4) Some confusing points are in supplement figure 1 and 2: similar with comment #(3), they have different VIP scores in different subplots. Furthermore, heatmaps in supplement figure 1D and 2D look very strange. Those heatmaps seem to be binary. What information do they express?

Response:

We truly appreciate the reviewer for pointing out the issue with supplementary Figures 1D and 2D. We have remade these two figures such that their parameters are consistent with those in Figure 2C. Also, we have guided readers to find the details regarding the data analysis for Figure 2C and Supplementary Figures 1D, 2D in our Methods section.

Reviewer #3 (Remarks to the Author): Expert in glioma surgery

In their manuscript entitled “Using arterial-venous analysis to characterize cancer metabolic consumption in patients” Xiong and colleagues provide insight into the metabolic properties of gliomas through a comparison of metabolites within arterial and venous blood associated with human gliomas. I have been a reviewer for over one hundred manuscripts in my career. This is easily one of the most interesting and bold experimental designs I have encountered. I have a strong bias towards the publication of this paper for this reason alone. I think the data are extraordinarily valuable because of the rigor of the experimental design and they are hard to argue with. I can think of no better way to study the key elements of human glioma metabolism. Therefore, I think the results will be of high value to the field.

Response

We thank the reviewer for this encouraging feedback about our study. We hope the method that we established here will benefit our scientific community and can be applied to other cancer studies in patients or animal models.

In glioma biology and in the care of glioma patients the IDH mutation is central. I was surprised in reading this manuscript- which certainly contains IDH mutant and wild-type tumors that the authors did not evaluate the upstream and downstream concentrations of 2HG. I view this as an essential internal control for their methodology. As there is no physiologic source of 2HG in the body, 2HG should be low in arterial blood and high in venous blood in an IDH mutant tumor. In IDH wild-type tumors the concentration of 2HG should not vary between arterial and venous blood- and may not be detectable. To me, including a detailed analysis of 2HG in the collected samples is essential. Without this, I think the authors are missing a key control. With 2HG measurements that make sense, I think the authors will greatly improve the acceptance of the other metabolic results they provide.

Response

This is a very good question. In our previous measurement, we found that it was extremely difficult for us to obtain “clean” signals of 2HG from our measurement with QTRAP5500 (AB SCIEX QTRAP 5500 LC/triple quadrupole mass spectrometer) because it was always fused with the peak of another unknown metabolite in plasma. They had very close retention times (see the figure below). As a result, we did not include 2HG in our previous manuscript. We have now performed additional measurements that permitted us to add analysis of 2HG to the revised manuscript as Supplementary Figure S3.

We went back and carefully re-analyzed all of our original data and observed a tendency that the 2HG concentrations in glioma venous samples from patients with grade II and III gliomas (5 of 6 patients, oligodendroglioma and astrocytoma) were much higher than those in other gliomas (e.g., GBM) after their values were normalized to the samples from a peripheral vein in the same patients (*Supplementary Figure 4*, also see it below)

To further confirm the results, we used QTOF (Agilent 6550 iFunnel LC/quadrupole-time of flight mass spectrometer) to measure 2HG from these samples. We observed that 2HG signals in 4 of 6 patients with grade II and III gliomas were significantly higher than those in other patients (*Supplementary Figure 5*). It is a pity that we did not perform genomic sequencing of gliomas from all of these patients and identify the mutations in these gliomas, so we don't know which patient(s) have *IDH1/2* mutations. Somatic mutations in isocitrate dehydrogenases1 (*IDH1*) were first described in 2008 in 12% of glioblastomas (Parsons et al, 2008). *IDH1/2* are commonly mutated genes in grade II and grade III gliomas, with incidences of >75% (Kim et al., 2010, Hartmann et al., 2009). Our analysis of 2HG, although not ideal, reasonably reflects the metabolism of grade II and III gliomas in patients. We have included all of these results in *Supplementary Figures 3–5* in our revised manuscript.

Parsons DW, Jones S, Zhang X, et al. An integrated genomic analysis of human glioblastoma multiforme. *Science*. 2008;321(5897): 1807–1812.

Kim YH, Nobusawa S, Mittelbronn M, et al. Molecular classification of low-grade diffuse gliomas. *Am J Pathol*. 2010;177(6):2708–2714.

Hartmann C, Meyer J, Bals J, et al. Type and frequency of IDH1 and IDH2 mutations are related to astrocytic and oligodendroglial differentiation and age: a study of 1,010 diffuse gliomas. *Acta Neuropathol*. 2009;118(4):469–474.

Reviewers' comments:

Reviewer #2 (Remarks to the Author):

In the revised manuscript, the authors significantly improve the readability of the manuscript, especially for the concept of "VIP" metric and the structure of their main figures. However, there are still one minor issue which needs to be addressed before publication in Nature Communications:

1. In "Data Analysis" section of the Methods, the author add lots of details related to the calculation of VIP scores and construction of figures. However, for the concept of VIP scores, the content is insufficiently mathematically defined for statisticians to rigorously evaluate, nor does it give an intuitive explanation for readers who are not familiar with it. For the content of figures, the explanation for figure 2B is mixed with that for figure 2C. I'd suggest to move those contents related to figures to main text or figure captions to make it more accessible to readers.

2. Please add the full name and definition of TIC. I guess it is concentration of each metabolite?

Reviewer #3 (Remarks to the Author):

To me, the author's response to my suggestion to include an analysis of plasma concentration of 2HG upstream (within the arterial supply of the tumor) and 2HG downstream (in the venous drainage of the tumor) is not sufficient. There are well documented methods for the use of mass spectrometry to detect 2HG (Santagata et al PNAS July 29, 2014 111 (30) 11121-11126). It appears that the method the authors have chosen another method using the QTRAP5500 does not separate 2HG from a similarly sized confounding metabolite.

A bigger problem is that the authors cannot provide even basic genetic characterization of the tumors they've analyzed. Basic molecular characterization of gliomas is the standard of care in clinical practice and can certainly not be avoided in a research setting. If the authors no longer have tissue or samples from these patients, I would suggest a prospective experimental design where then can collect arterial and venous blood, as well as tissue for basic genetic analysis: IDH status, ATRX status, 1-19q status. To me a basic experiment showing that 2HG is present in very low concentration upstream and high concentration downstream is essential. Without this it's very hard for me to understand the metabolic results being presented here. That said, I still think this manuscript represents great science- it just needs to be strengthened to ensure the data are interpretable within the current diagnostic framework used to describe gliomas.

Manuscript Number: NCOMMS-19-26465-T

Using arterial-venous analysis to characterize cancer metabolic consumption in patients

A summary of our revision:

We appreciate the thoughtful remarks from all of the reviewers. To summarize, in our revision, we have made these modifications:

- (1) We have rewritten part of the Data Analysis portion of the Methods section and the related description in the main text;
- (2) We have provided an explanation of TIC (Total Ion Chromatogram) in the main text;
- (3) We have added the staining results of ATRX, P53 and IDH1 in Table 1;
- (4) We marked the patients with *IDH1* mutation in Figure 5c and Supplementary Figure 2;
- (5) We performed a new experiment measuring D/L-2-HG and these results were added in Figure 5d and Supplementary Figure 2;
- (6) We added the details of D/L-2HG measurement in the Methods section;
- (7) In the Discussion, we have discussed the caveats of our experimental design.

Reviewer #2 (Remarks to the Author):

In the revised manuscript, the authors significantly improve the readability of the manuscript, especially for the concept of “VIP” metric and the structure of their main figures. However, there are still one minor issue which needs to be addressed before publication in Nature Communications:

1. In “Data Analysis” section of the Methods, the authors add lots of details related to the calculation of VIP scores and construction of figures. However, for the concept of VIP scores, the content is insufficiently mathematically defined for statisticians to rigorously evaluate, nor does it give an intuitive explanation for readers who are not familiar with it. For the content of figures, the explanation for figure 2B is mixed with that for figure 2C. I’d suggest to move those contents related to figures to main text or figure captions to make it more accessible to readers.

Response:

We thank the reviewer for the suggestion. We have rewritten the whole Data Analysis portion of the Methods. Now it is much more concise. We have also cited two related pieces of literature for reference.

Data analysis

Integrated chromatogram peaks of each metabolite were analyzed with MultiQuant software (AB Sciex). The ion intensity was calculated by normalizing single ion values against the total ion value of the entire chromatogram (i.e. TIC or Total Ion Chromatogram). The data matrix was input into SIMCA-P software (Umetrics) by mean-centering and Pareto scaling for subsequent analysis so that the model fitting would not be biased by concentrations and variations of different metabolites. Both unsupervised and supervised multivariate data analysis approaches including principal component analysis (PCA), hierarchical clustering, and partial least squares discriminant analysis (PLS-DA) were performed using Metaboanalyst 4.0 (Chong *et al.*, 2018). We performed feature selection in PLS-DA to identify metabolites that maximize separation between the venous and arterial groups by rotating the PCA components. The importance of a metabolite in the model is measured by the Variable Importance in Projection (VIP) score. The VIP score of a metabolite is calculated as a weighted sum of the squared correlations between this metabolite and the derived PLS-DA components. Each weight corresponds to the percentage variation of the response variable, i.e., gA and gV, explained by a PLS-DA component. Intuitively, the VIP score of a metabolite indicates its intensity of association with the PLS-DA components that best distinguish the gA and gV groups. By definition, the average of squared VIP scores equals 1, and by convention a VIP score of greater than 1 is used as a criterion for variable selection (Chong & Jun, 2005). Thus, metabolites with a VIP score > 1 were reported.

References:

1. Chong J, Soufan O, Li C, Caraus I, Li S, Bourque G, Wishart DS, Xia J. MetaboAnalyst 4.0: towards more transparent and integrative metabolomics analysis. *Nucleic Acids Res.* 2018 46(W1):W486-W494. doi: 10.1093/nar/gky310.
2. Chong IG, Jun CH. Performance of some variable selection methods when multicollinearity is present. *Chemometrics and Intelligent Laboratory Systems* 78 (2005) 103–112

2. Please add the full name and definition of TIC. I guess it is concentration of each metabolite?

Response:

We have added the definition of TIC in our revision: A TIC (Total Ion Chromatogram) is a chromatogram created by summing up intensities of all metabolites that we obtained from the same scan.

Reviewer #3 (Remarks to the Author):

To me, the author's response to my suggestion to include an analysis of plasma concentration of 2HG upstream (within the arterial supply of the tumor) and 2HG downstream (in the venous

drainage of the tumor) is not sufficient. There are well documented methods for the use of mass spectrometry to detect 2HG (Santagata et al PNAS July 29, 2014 111 (30) 11121-11126). It appears that the method the authors have chosen another method using the QTRAP5500 does not separate 2HG from a similarly sized confounding metabolite.

A bigger problem is that the authors cannot provide even basic genetic characterization of the tumors they've analyzed. Basic molecular characterization of gliomas is the standard of care in clinical practice and can certainly not be avoided in a research setting. If the authors no longer have tissue or samples from these patients, I would suggest a prospective experimental design where then can collect arterial and venous blood, as well as tissue for basic genetic analysis: IDH status, ATRX status, 1-19q status. To me a basic experiment showing that 2HG is present in very low concentration upstream and high concentration downstream is essential. Without this it's very hard for me to understand the metabolic results being presented here. That said, I still think this manuscript represents great science- it just needs to be strengthened to ensure the data are interpretable within the current diagnostic framework used to describe gliomas.

Response:

The suggestion from the reviewer is very constructive. This is a very exploratory and high-risk study regarding tumor metabolism, we admit that there are caveats in our experimental design. We should have saved some tumor tissues from these patients and done sequencing. We went back to the database from the hospital and found further diagnostic information regarding these patients. Fortunately, we obtained the staining results for some of the patients, though samples from the hospital are not always sent for further staining. Glioma tissues from four of these patients have IDH1 mutations. We have added this information in our Table 1 (ATRX, P53 and IDH1). Interestingly, we found that three samples from patients with IDH1 mutants had the highest 2HG signal in venous samples (we have marked them in Figure 4c).

Patient No.	Sex	Age	Glioma type	IDH1	ATRX	P53	WHO type
1	M	58	Astrocytoma	-	-	-	II
2	F	21	Glioblastoma	-	n.a.	n.a.	IV
3	F	53	Astrocytoma	n.a.	n.a.	n.a.	II
4	F	27	Astrocytoma	n.a.	n.a.	n.a.	III
5	M	53	Glioblastoma	-	-	-	III-IV
6	F	60	Gliosarcoma	n.a.	n.a.	n.a.	IV

7	F	57	Glioblastoma	n.a.	n.a.	n.a.	IV
8	F	46	Glioblastoma	n.a.	n.a.	n.a.	III-IV
9	M	61	Oligodendroglioma	+	+	+	III
10	M	44	Oligodendroglioma	+	+	-	III
11	M	53	Astrocytoma	+	-	-	II
12	M	42	Astrocytoma	+	-	+	II
13	F	61	Glioblastoma	n.a.	n.a.	n.a.	IV

In addition, we used a method (see it below) described in a previous study (Struys, et al., 2004) to measure D-2-HG and L-2-HG in arterial, venous, and peripheral samples from these patients with *IDH1* mutation (based on staining results, patients No. 9–12, see Table 1). We were so excited to observe that D-2-HG was significantly higher in venous samples than in arterial samples from the same patients (See the figure below. It has been added in Figure 5d, paired *t*-test, *p*-value = 0.013. It was added into Figure 4d). We should also note that we observed the D-2-HG concentration was very low in all peripheral plasma samples (<1 μ M), which is also consistent with the measurement from previous study of D-2-HG in peripheral venous samples (Wang *et al.*, 2013).

Measurement of D-2-hydroxyglutarate (D-2-HG) from plasma

Metabolites were extracted with 80% methanol-water solution from 25 μ L plasma from patients in a tube. A SpeedVac was used to dry the extract into a pellet. To the pellet was added U13C-D/L-2-hydroxyglutarate (internal standard, Cambridge isotope laboratories, 10 nG in 10 μ L acetonitrile), then the mixture was then dissolved in 90 μ L freshly mixed 80% acetonitrile/20% acetic acid plus 50 mg/mL diacetyl-L-tartaric anhydride (DATAN, Acros Organics). The solution thus obtained was sonicated and warmed up to 75°C for 30 min. Samples were cooled to room temperature and centrifuged. The supernatant was dried with a SpeedVac, and the pellet was reconstituted into 1.5 mM ammonium formate aqueous solution with 10% acetonitrile (100 μ L). LC/MS analysis was performed on an AB Sciex 5500 QTRAP liquid chromatography/mass spectrometer (Applied Biosystems SCIEX) equipped with a vacuum degasser, a quaternary pump, an autosampler, a thermostatted column compartment, and a triple quadrupole/iontrap mass spectrometer with electrospray ionization interface, and controlled by AB Sciex Analyst 1.6.1 Software. Waters Acquity UPLC HSS T3 column (150mM \times 2.1mM, 1.8 μ M) column was used for separation. Solvents for the mobile phase were 1.5 mM ammonium formate aqueous (pH 3.6 adjusted with formic acid (A), and pure acetonitrile (B). The gradient

elution was: 0–12 min, linear gradient 1–8% B and 12–15 min, 99% B, then the column was washed with 99% B for 5 min before reconditioning it for 3 min using 1% B. The flow-rate was 0.25 mL/min and the column was operated at 35°C. Multiple reaction monitoring (MRM) was used to check 2-hydroxyglutarate-diacetyl tartrate derivatives: 363/147 (CE: -14V); 368/152 (internal standard, CE: -14V).

Reference:

1. Struys EA, Jansen EE, Verhoeven NM, Jakobs C. Measurement of urinary D- and L-2-hydroxyglutarate enantiomers by stable-isotope-dilution liquid chromatography-tandem mass spectrometry after derivatization with diacetyl-L-tartaric anhydride. *Clin Chem*. 2004 50(8):1391-5.
2. Wang JH *et al*. Prognostic significance of 2-hydroxyglutarate levels in acute myeloid leukemia in China. *Proc Natl Acad Sci U S A*. 2013 110(42):17017-22.

REVIEWERS' COMMENTS:

Reviewer #2 (Remarks to the Author):

The authors have addressed my concerns.

Reviewer #3 (Remarks to the Author):

The response to my suggestions is very nicely constructed. Ideally it would be great to see the observation that 2HG levels in venous samples greatly exceed that of plasma or arterial samples in a larger series of patients but I am quite satisfied by the data presented and support publication at this point.

REVIEWERS' COMMENTS:

Reviewer #2 (Remarks to the Author):

The authors have addressed my concerns.

Reviewer #3 (Remarks to the Author):

The response to my suggestions is very nicely constructed. Ideally it would be great to see the observation that 2HG levels in venous samples greatly exceed that of plasma or arterial samples in a larger series of patients but I am quite satisfied by the data presented and support publication at this point.

We greatly appreciate the reviewer's positive comments on our revised manuscript.